# Assessment of the Chemical Profile and Potential Medical Effects of a Flavonoid-Rich Extract of *Eclipta prostrata* L. Collected in the Central Highlands of Vietnam

**DOI:** 10.3390/ph16101476

**Published:** 2023-10-16

**Authors:** Thi Kim Phung Phan, San-Lang Wang, Quang Vinh Nguyen, Tu Quy Phan, Tan Thanh Nguyen, Thanh Tam Toan Tran, Anh Dzung Nguyen, Van Bon Nguyen, Manh Dung Doan

**Affiliations:** 1Faculty of Medicine and Pharmacy, Tay Nguyen University, Buon Ma Thuot 630000, Vietnam; ptkphung@ttn.edu.vn; 2Faculty of Pharmacy, University of Medicine and Pharmacy at Ho Chi Minh City, Ho Chi Minh City 700000, Vietnam; 3Department of Chemistry, Tamkang University, New Taipei City 25137, Taiwan; 4Life Science Development Center, Tamkang University, New Taipei City 25137, Taiwan; 5Institute of Biotechnology and Environment, Tay Nguyen University, Buon Ma Thuot 630000, Vietnam; nqvinh@ttn.edu.vn (Q.V.N.); nadzung@ttn.edu.vn (A.D.N.); nvbon@ttn.edu.vn (V.B.N.); 6Department of Science and Technology, Tay Nguyen University, Buon Ma Thuot 630000, Vietnam; phantuquy@ttn.edu.vn; 7School of Chemistry Biology and Environment, Vinh University, Vinh City 43100, Vietnam; nguyentanthanhvn@gmail.com; 8Faculty of Pharmacy, Dong A University, Da Nang City 50000, Vietnam; toanttt@donga.edu.vn

**Keywords:** *Eclipta prostrata* L., UHPLC, flavonoids, anti-oxidant, anti-acetylcholinesterase

## Abstract

*Eclipta prostrata* L. (EPL), a medicinal plant, is widely utilized in the central highlands of Vietnam. This study aims to assess the chemical profile and potential medical effects of an EPL extract rich in flavonoids. A total of 36 secondary metabolites were identified from the EPL extract through GC-MS and UHPLC-UV analysis. Among them, 15 volatile compounds and several phenolic and flavonoid chemicals, including salicylic acid, epicatechin gallate, isovitexin, and apigetrin, were reported in EPL extract for the first time. This herbal extract demonstrated moderate inhibition against α-amylase and α-glucosidase, and high anti-oxidant and anti-acetylcholinesterase activities (IC_50_ = 76.8 ± 0.8 μg/mL). These promising attributes can be likely attributed to the high levels of major compounds, including wedelolactone (**1**), chlorogenic acid (**3**), epicatechin gallate (**6**), salicylic acid (**8**), isovitexin (**9**), apigetrin (**11**), and myricetin (**12**). These findings align with the traditional use of EPL for enhancing memory and cognitive function, as well as its potential benefits in diabetes management. The results of the molecular docking study reveal that the major identified compounds (**1**, **6**, **9**, and **11**) showed a more effective acetylcholinesterase inhibitory effect than berberine chloride, with good binding energy (DS values, −12.3 to −14.3 kcal/mol) and acceptable values of RMSD (1.02–1.67 Å). Additionally, almost all the identified major compounds exhibited good ADMET properties within the required limits.

## 1. Introduction

The research and use of medicinal plants for treating various diseases have gained significant attention and stand as a top priority for researchers worldwide. This trend is driven by the growing interest in natural and traditional remedies, as well as the need to discover new therapeutic options [1,2,3]. Vietnam, a tropical country, is blessed with a rich diversity of flora and fauna, including a vast array of medicinal plants. This study focuses on investigating the chemical profiles and some medical effects of *Eclipta prostrata* Linn. (EPL), an herbaceous botanical species renowned not only in Vietnam but also embraced across Asian and African traditional healing practices.

EPL is widely used in folk medicine to treat various diseases, namely, hepatic and renal diseases, respiratory disorders, skin diseases, fever, diabetes, hypertension, lacerations and injuries, and to enhance memory and cognitive function [4,5]. Due to its wide range of uses, a large number of studies have been conducted to isolate and characterize its biologically active constituents and determine its pharmaceutical potential. The crude extracts and individual compounds isolated from EPL have been reported to exhibit promising pharmacological properties, such as anti-inflammatory [6], anti-oxidant [7,8,9], anticancer [10], antimicrobial [11], hair growth promotion [12], hepatoprotective [13], anti-diabetic [14,15] and neuroprotective effects [16].

In previous studies, a variety of chemical substances have been extracted and identified from *E. prostrata*, including cardiac glycosides, alkenynes, alkaloids, flavonoids, steroidal alkaloids, coumestans, lipids, steroids, saponins, phytosterol, polyacetylene, triterpenes, and other compounds [17,18]. Among these, the main compounds include flavonoids, steroids, thiophenes, coumarins and triterpenoids [4]. Numerous studies have demonstrated that flavonoids are the primary constituents of EPL and exert a significant impact on biological activities [13,19,20]. One strategy that has gained attention in addressing neurodegenerative diseases like Alzheimer’s involves dietary interventions featuring flavonoid-rich foods [21]. In addition, a number of studies have demonstrated the potential health benefits of natural flavonoids in treating diabetes [22,23,24].

This recognition stems from the diverse pharmacological attributes associated with flavonoids, including anti-oxidant, anti-inflammatory, and potential glucose-regulating effects. However, studies on the chemical profile and potential medical effect of EPL extract, which is rich in flavonoids, remain limited. Several flavonoids have been tentatively identified within the plant, such as wedelolactone, myricetin, quercetin, kaempferol, apigenin [17,25], luteolin, apigenin, luteolin 7-*O*-*β*-D-glucopyranoside [26], luteolin 7-O-glucoside, demethylwedelolactone, wedelolactone, luteolin, demetylwedelolactone sulfate, luteolin sulfate, and apigenin sulfate [20]. Therefore, studying the extraction process to obtain flavonoid-rich extracts from EPL and investigating their biological activities are of scientific significance and necessity.

As global concerns regarding diabetes mellitus and Alzheimer’s escalate, the need for affordable and efficacious treatments becomes increasingly essential [27,28]. This article presents an investigation into optimizing experimental conditions to obtain an extract from EPL that is rich in flavonoids. After that, we identify the chemical composition and potential medical applications of EPL through assessments encompassing anti-oxidant, anti-glucosidase, anti-amylase, and anti-acetylcholine activities. In doing so, this research highlights the extract’s role in the treatment of diseases like diabetes and Alzheimer’s.

## 2. Results and Discussion

### 2.1. Extraction Optimization of Flavonoids from Eclipta prostrata L. Using Response Surface Methodology

#### 2.1.1. Fitting the Response Surface Models

The Box–Behnken design was employed to define the relationship between the response features and procedure changes, as well as to determine the optimal extraction conditions. The total flavonoid content of the EPL extracts was optimized as the response variable. The experimental design and the representation of the three independent variables are described in Table 1.

In this research, a quadratic multinomial sample was chosen and found to be well-suited for all three independent variables and the response variable, as proposed by the employed software. The empirical relationship between the dependent (Y) and three independent (X) variables is described by the following quadratic polynomial equation (Equation (1)):Y = 172.49 + 0.75X_1_ + 0.96X_2_ + 0.55X_3_ – 0.45X_1_X_2_ – 0.48X_1_X_3_ – 0.23X_2_X_3_ – 2.74X_1_^2^ – 0.82X_2_^2^ – 0.84X_3_^2^(1)
where Y represents the total flavonoid content; and X_1_, X_2_ and X_3_ denote extraction temperature, extraction time, and extraction pressure, respectively.

A minus sign in the formula describes an antagonistic impact of the changes, while a positive or plus sign indicates a synergistic impact of the changes. The coefficients of the response surface test samples were verified through an analysis of variance (ANOVA) for the quadratic multinomial sample, as reported in Table 2. The model F-value of 51.97 indicates that the model was highly significant at *p* < 0.0001, which indicates that the sample is notable and that there is only a 0.010 percentage chance that this very large F-value could occur due to noise. The F-value (0.1569) for the lack of fit was insignificant (*p* = 0.9200 >0.05), confirming the adequacy of the model. The high values of R^2^ (0.9853) and adjusted R^2^ (0.9663) indicate a strong correlation between the response and independent variables, signifying that the model accurately represents the actual relationship. Moreover, the low coefficient of the variation value (CV = 0.2035) clearly indicates high precision and good reliability of the experimental values.

#### 2.1.2. Response Surface Analyzed Process

According to the analysis procedure of the ANOVA program, for all terms in the sample of quadratic polynomial coefficients, large F-values and small *p*-values describe more significant effects on changes in the corresponding responses. The utilized software creates a 3D surface plot of the fitted polynomial regression formula to better visualize the interactive influence of independent changes on the response.

In this work, extraction temperature, extraction time, and pressure were the main extraction conditions that are important in achieving the highest flavonoid content. This section elaborates on how these factors play a role in the extraction of potent anti-oxidants. A 3D sample plot is shown below, illustrating corresponding sample surface area plots. These plots involve varying three variables within the test area during the investigation while keeping the other factors at their central order (zero levels).

The impacts of their changes on the total flavonoid content (TFC) are illustrated in Figure 1a–c. The quantity of TFC extracted from EPL ranged between 166.72 and 173.05 mg QE (quercetin equivalents)/g sample extract. The highest TFC was found in test run no. 13. TFC was significantly impacted at *p* < 0.050 by all three linear elements (X_1_, X_2_ and X_3_), interaction parameters (X_1_X_2_ and X_1_X_3_) and all quadratic parameters (X_1_^2^, X_2_^2^ and X_3_^2^) (described in Table 2).

The surface zone plot in Figure 1a illustrates the influence of extraction temperature and extraction time on TFC while maintaining a constant pressure of 100 bar. The extraction time is an important factor influencing extraction efficiency because it is time-consuming for EPL to release flavonoids into the solvent. At 50 °C, the quantity of TFC was lowest when the extraction time was 30 min, and this figure rose gradually as the extraction time was extended from 30 to 60 min. The temperature of the extraction process also affected the TFC in its interaction with the time of extraction. It was found that raising the extraction heat from 50 to 65 °C resulted in higher TFC. However, a continuous rise in temperature led to an inversely proportional impact, particularly as the extraction duration increased. The reason for this trend could be that flavonoid substances are endangered by the oxidation procedure or the degradation process [29]. The greatest quantity of TFC was obtained at temperatures of 60–65 °C and an extraction period of 50–60 min.

Figure 1b shows the influence of heat versus pressure at an extraction time of 45 min. The 3D response surface indicates a complex interaction effect between extraction temperature and extraction pressure. While the increase in the pressure at low levels of temperature (50–55 °C) led to a rise in TFC, at higher temperatures, the pressure had a minor effect on TFC, with only a slight increase when pressure changed from 80 to 120 bar. The maximum TFC could be acquired from EPL with pressure and extraction temperature levels of roughly 100–110 bar and 55–65 °C, respectively.

The response signal outer surface zone plot as a feature of the extraction period versus pressure at a constant heat of 60 °C is shown in Figure 1c. The combined effect of extraction time and pressure is not very prominent. At low pressure, an increase in the period of extraction leads to an increase in TFC. However, at high pressure, the increased time does not significantly increase TFC. The outer surface zone plots illustrate that a higher TFC could be achieved when the process is carried out under rising pressure for a suitable extraction period.

Overall, considering the interaction of factors, TFC extraction efficiency tends to be moderate at both low and high levels of extraction pressure and temperature. At the middle levels of these two factors, TFC efficiency showed an increasing trend. The highest concentration of TFC was observed at the medium temperature (55–65 °C) and pressure (100–110 bar) ranges of the extraction process. The increase in extraction time generally resulted in higher TFC, but within the middle range of temperature and pressure, this trend was not as clear, and the optimal extraction time appears to be around 50–55 min.

#### 2.1.3. Optimization and Model Verification

The purpose of this study was to investigate the conditions that generated the greatest yield of total flavonoid content. The final outcome of the dual optimization, which employs the desirability feature technique, proposes optimal extraction conditions for EPL of 61 °C for 53 min at 105 bar of pressure. These conditions resulted in the highest flavonoid content, which was 172.83 ± 0.25 mg QE/g (n = 3). The results imply that the experimental test values were consistent with the expected values.

Previous studies have employed various methods to extract flavonoids from EPL, including Soxhlet apparatus extraction, ultrasonic extraction, and immersion methods. These methods yielded TFC values ranging from 10 to 90 mg QE/g of dry weight [7,8,30]. However, this current research utilized pressurized liquid extraction (PLE) and achieved a significantly higher TFC of 172.83 ± 0.25 mg QE/g. This result suggests that PLE is a more efficient extraction method for obtaining flavonoids from EPL.

Moreover, in terms of optimizing the extraction process, prior studies have utilized methods like ultrasonic-assisted extraction and microwave-assisted extraction to optimize specific compounds such as wedelolactone, several phenolic compounds and total saponin content [31,32,33]. However, this study marks the pioneering application of the PLE method to extract TFC compounds from EPL samples collected in the Tay Nguyen region. Furthermore, by employing PLE, the research aims to establish an environmentally friendly and sustainable approach to obtaining valuable components from the collected plant samples.

### 2.2. Chemical Profiles of Eclipta prostrata L. Extract Rich in Flavonoid Content

The optimal extract was analyzed for its phytochemical compositions via UHPLC and GC-MS methods.

It was demonstrated that the herbal EPL is abundant in phenolic ingredients. In this study, the UHPLC-UV method was used to determine the chemical components and their concentrations. When the extract was analyzed using UHPLC-UV, a total of 15 chemicals (**1**–**15**) were found and are listed in Table 3.

According to Vietnamese Pharmacopoeia (Edition, 2018) [34], Wedelolactone was identified as a qualitative marker for EPL, and the Chinese Pharmacopoeia (Edition, 2020) [35] stipulated that the content of wedelolactone in the aerial parts of the plant should be higher than 0.04% when analyzed by HPLC. In this study, Wedelolactone (**1**) was found as the main chemical in the EPL extract, constituting approximately 0.08% of the whole plant. This finding suggests that it could play a significant role in EPL’s overall therapeutic properties. Wedelolactone has been reported to possess numerous medicinal effects, such as anti-oxidant [36], anti-diabetic [22], and anti-acetylcholinesterase (anti-AchE) [25] activities, as well as anti-inflammatory activity mediated via inhibition of the NF-kappa-B pathway [37] or through the trypsin inhibitory effect [38]. In addition, it acts as a growth suppressor independently of NFκB and androgen receptors, reducing the growth of MDA-MB-231 breast cancer cells [39]. It inhibits the growth of prostate cancer cells by triggering apoptosis in cancer cells and specifically targeting PKCε [40], and shows inhibitory effects on the proliferation and migration of head and neck squamous cancer cells. In addition, other substances such as chlorogenic acid (**3**), myricetin (**12**), quercetin (**13**), kaempferol (**14**), apigenin (**15**) were also were also identified as constituents present in the EPL extract, consistent with findings from previously published studies [17,25].

However, in this research, salicylic acid (**8**), which is a phenolic component, accounts for the highest level in this herb, reaching 7912.08 ± 0.03 μg/g. Salicylic acid is widely distributed in many plants and has been used to relieve pain, reduce fever and prevent heart attack and stroke. It also plays a vital role in many skin products used to treat acne, psoriasis, photoaging, and wounds from burns due to its anti-inflammatory effect [41,42]. Interestingly, this is the first time that salicylic acid has been reported as a constituent of EPL. This finding may contribute to explaining the anti-inflammatory activity of EPL that has been reported previously [6,43]. Additionally, epicatechin gallate (**6**), isovitexin (**9**), rutin (**10**) and apigetrin (**11**), which are present in high concentrations, were found in ELP for the first time.

Figure 2 depicts the chemical structures of these phenolics, and the UHPLC fingerprinting is displayed in Figure 3.

Furthermore, the GC-MS method was applied to detect the volatile compounds present in the extract of EPL. A total of 21 volatile ingredients (**16**–**36**) were identified and are listed in Table 4.

In the table, the compounds are arranged in the order of their detected peaks based on the recorded retention times. Among them, 3′,4′,5′,5,6,7-hexamethoxyflavone had the highest content at 77.97 ± 0.05%. Additionally, other components with relatively high contents were also found, such as 13,14-Dihydro-15(R)-prostaglandin E1 at 1.18 ± 0.02%, Hexadecanoic acid at 3.01 ± 0.02%, 9,12-Octadecadienoic acid (Z,Z)- at 3.59 ± 0.04%, all-trans-5,8-Epoxyretinoic acid at 2.16 ± 0.04%, and Pregna-5,16-dien-20-one, 3.21-bis(acetyloxy)-, (3β)- at 2.33 ± 0.02%. Figure 4 depicts the chemical structures of these volatile chemicals, and their GC-MS profiles are displayed in Figure 5. Fifteen compounds were initially detected from EPL plant extracts collected in Dak Lak. Moreover, according to the chemical composition determined by the GC-MS method, some of the detected compounds are similar to those reported in previous publications, such as 9,12-Octadecadienoic acid (Z,Z)-; Hexadecanoic acid; Pentadecane, 2-methyl- [44,45]; and Germacrene D [4,46]. The variation in chemical composition of EPL plants demonstrates the impact of climate and soil on active component production. The findings indicate that EPL trees in Dak Lak have the potential to be used as a source of active compounds with beneficial biological benefits.

### 2.3. Assessment of Novel and Potential Medical Effects of Eclipta prostrata L. Extract

The medicinal effects of the optimized EPL extract were examined, including its anti-oxidant, anti-diabetic, and anti-Alzheimer activities. Table 5 shows the bioactivities of these extracts as IC_50_ values (half-maximal inhibitory concentration of samples). The effects of these bioactivities are reported as IC_50_ values (Inhibition concentration of 50%), where a lower value indicates greater activity.

Free radicals can assault many kinds of cells in the body, causing oxidative stress which contributes to various chronic diseases like cancer, neurodegeneration, diabetes, and cardiovascular and inflammatory disease [47,48]. Hence, researching its anti-oxidant activity is useful for evaluating the potential applications of an herb for medicinal purposes. Popular methods for measuring anti-oxidant ability include scavenging of DPPH (2,2-diphenyl-1-picrylhydrazyl—a free radical) and ABTS (2,2′-azino-bis(3-ethylbenzothiazoline-6-sulfonic acid)) cation radicals. Thus, in this study, DPPH and ABTS radical scavenging capabilities were used to assess the anti-oxidant ability of methanolic extracts of EPL. As shown in Table 5, IC_50_ values of 130.2 ± 1.0 μg/mL and 38.4 μg/mL were obtained for the extract and ascorbic acid, respectively, in the reduction of stable free radical DPPH. In addition, the EPL extract exhibited lower anti-oxidant ability, with an IC_50_ value of 79.9 ± 0.9 μg/mL in comparison with that of ascorbic acid (23.0 ± 0.4 μg/mL) in the ABTS assay. In comparison, the anti-oxidant activity of EPL was found to be lower than that of vitamin C by 2 to 10 times in some studies [7,8,10]. In consequence, EPL may be suggested as a potential source for anti-oxidant compounds.

Diabetes mellitus is a significant global health concern that has been on the rise in recent decades. According to the World Health Organization (WHO), diabetes mellitus is a chronic, metabolic disease characterized by elevated levels of blood glucose, which can lead to severe complications affecting various organs and systems in the body, including cardiovascular diseases (heart attacks, strokes), kidney damage (nephropathy), eye problems (retinopathy), nerve damage (neuropathy), and foot complications [49]. Globally, there are 537 million diabetic patients, with 75% of adults with diabetes living in low- and middle-income countries, and 6.7 million deaths due to diabetes were reported in 2021 by the International Diabetes Federation (IDF) [27]. Thus, access to affordable and effective treatment is critical for people living with diabetes. The search for herbal treatments for diabetes has been ongoing for many years as traditional medicines often contain bioactive compounds that could potentially help manage blood glucose levels. In this research, the EPL extract was assessed for its anti-diabetic potential by evaluating its inhibitory activity on the two major enzymes involved in blood sugar increase, α-glucosidase [14] and α-amylase [15]. In these assays, the samples exhibited moderate enzyme inhibition activity against α-glucosidase (IC_50_ value: 223.1 ± 1.3 μg/mL) and α-amylase (IC_50_ value: 429.9 ± 2.1 μg/mL). The ability of EPL collected in Dak Lak to counter diabetes was greater than that observed in several previous studies [50,51]. Some components present in EPL, like wedelolactone [22], isovitexin [23], rutin [24], and chlorogenic acid [52], have been reported to possess the ability to inhibit α-glucosidase and α-amylase. Hence, further research aimed at identifying the specific bioactive compounds responsible for these activities should be conducted. Once identified, it may be possible to improve the anti-diabetic activity of EPL, by identifying and isolating the compounds that are responsible for the observed α-glucosidase and α-amylase inhibition.

Cholinesterase inhibitors are indeed a mainstay of pharmacological treatment for Alzheimer’s disease. These medications work by inhibiting the enzymes AchE and/or butyrylcholinesterase, which break down the neurotransmitter acetylcholine [28,53]. Thus, to evaluate the potential of EPL in Alzheimer’s disease treatment, assays testing the resistance of acetylcholinesterase were carried out. The extract exhibited anti-AchE activity, with IC_50_ values of 76.8 ± 0.8 μg/mL. These values were significantly lower than those of berberine chloride (301.1 ± 1.2 μg/mL), which was used as standard in the assay, indicating a higher anti-AchE potential of the EPL extract.

Salicylic acid and wedelolactone are two main compounds that accounted for more than 70 percent (*w*/*w*) of the total flavonoid content. To further understand the function of these main compounds, their IC50 values for anti-AChE activity were determined. The results (Table 5) show that salicylic acid and wedelolactone exhibited more effective AChE inhibitory activity, with IC50 values of 55.98 ± 0.12 μg/mL and 61.23 ± 0.21 μg/mL, respectively, compared with the IC50 value of 76.8 ± 0.8 μg/mL for the EPL extract. These results suggest that salicylic acid and wedelolactone play significant roles in the anti-AChE activity of the EPL extract. In addition, the high AChE inhibition potential may be due to the high levels of epicatechin gallate, myricetin, isovitexin, apigetrin [54,55] components in the EPL extract. These components also exhibit a high ability to inhibit AChE, as indicated by their IC_50_ values presented in Table 6.

EPL is a medicinal plant with a long history of traditional use in various Asian cultures. One of the notable traditional uses of *Eclipta prostrata* is to enhance memory and cognitive function [4,5]. Dae-Ik Kim demonstrated that a possible mechanism of the memory-enhancing effects of EPL may involve an increase in Ach and a decrease in oxidative stress, which may improve brain function [5]. Bioactivity research confirms that the extract has potential anti-oxidant, anti-diabetic, and anti-Alzheimer effects. Overall, our findings imply that EPL has the potential to be a natural source of acetylcholinesterase inhibitors, which could be useful in treating illnesses such as Alzheimer’s disease and other kinds of dementia [60].

### 2.4. Assessment of the Interactions and Energy Binding of Major Constituents of Eclipta prostrata L. Extract toward Acetylcholinesterase

In docking studies of an inhibitor ligand into targeting protein enzyme, the ligand is able to contact and bind to the enzyme at various sites on the enzyme (named binding sites, BSs). However, only the most stable intermolecular structure is chosen for investigation in detail [61]. A BS on Acetylcholinesterase (protein 1EEA) was identified using the site finder function of MOE and is illustrated in Figure 6. This BS was found to contain various prominent amino acids (GLN69, TYR70, VAL71, ASP72, GLN74, SER81, TRP84, ASN85, PRO86, TYR116, GLY117, GLY118, GLY119, TYR121, SER122, GLY123, SER124, LEU127, TYR130, GLU199, SER200, TRP233, TRP279, LEU282, PHE284, ASP285, SER286, ILE287, PHE288, ARG289, PHE290, PHE330, PHE331, TYR334, GLY335, HIS440, GLY441, TYR442, ILE444).

In the docking study, two forms of output data, the root mean square deviation (RMSD) and docking score (DS), were considered as important indicators in determining the significant binding (RMSD ≤ 2.0 Å) and effective inhibition (DS ≤ −3.20 kcal/mol) of an inhibitor towards the targeting enzyme [62,63]. Thus, these two indicators were used to analyze the interactions and predict the active inhibitory effects of the major compounds identified in the EPL extract.

As summarized in Table 7, nearly all the compounds showed low RMSD values in the range of 1.02–1.98 Å (lower than 2.0 Å), indicating the successful binding of all the ligands to the enzyme 1EEA with acceptable RMSD values. In virtual drug screening, a compound with a DS value less than −3.20 kcal/mol is proposed as an enzyme inhibitor [63]. As shown in Table 7, all the identified compounds bound to enzyme 1EEA with good energy binding (DS values in the range of −10.0 to −14.3 kcal/mol), indicating that they may be potential 1EEA candidates. Berberine chloride, a commercial inhibitor, was also analyzed for comparison and showed good energy binding to 1EEA (DS value, −12.1 kcal/mol). In the comparison based on DS values, six compounds (**8**, **28**, **31**, **33**, **34**, and **35**) exhibited weaker inhibitory effects against 1EEA (DS values ranging from −10.0 to −11.7 kcal/mol) than berberine chloride (DS values of −12.1 kcal/mol). Conversely, four other compounds (**1**, **6**, **9**, and **11**) showed higher 1EEA inhibitory activity (DS values, −12.3 to −14.3 kcal/mol) than berberine chloride (**37**). Generally, the order of 1EEA inhibitory effect of these tested compounds was as follows: **6** > **11** > **9** > **1** > **37** (berberine chloride) > **28** > **31** > **33** > **34** > **8** > **35**.

To gain a deeper understanding of the interaction between the most effective inhibitor compounds (**1**, **6**, **9**, and **11**) and berberine chloride (**37**), the detailed interaction at the BSs is presented and discussed (Figure 7). Epicatechin gallate (**6**) showed the best docking score against 1EEA (DS value of −14.3 kcal/mol) via interaction with two amino acids (SER81 and GLU199), creating three linkages with an H-donor. Of these, this ligand was found to be connected to SER81, generating an H-donor linkage at the distance and energy binding values of 2.68 Å and −2.4 kcal/mol, respectively. It also formed two H-donor linkages with GLU199 at distances ranging from 2.68 to 3.24 Å and energy binding values ranging from −2.1 to −5.6 kcal/mol. It was followed by apigetrin (**11**), which showed a strong binding energy, with a DS value of −13.7 kcal/mol by interacting with five amino acids (GLU199, ASP72, SER 286, PHE331, and PHE288) to create five linkages (3 H-donor, 1 H-pi, and 1 pi-H). Apigetrin (**11**) was found to bind to 1EEA with up to five linkages, while epicatechin gallate (**6**) interacted with this enzyme via only two linkages. However, the inhibition effect (based on binding energy, DS value) of ligand 2 is better than that of ligand 5. These results indicate that the inhibitory effect of a ligand may be independent of the number of interactions between the ligand and the enzyme [4]. The next two compounds, wedelolactone (**1**) and isovitexin (**9**), could also effectively bind to 1EEA with DS values of −12.3 and −13.4 kcal/mol, respectively. Wedelolactone (**1**) bound to 1EEA by interacting with GLY118, and two pi-H linkages were formed. Isovitexin (**9**) interacted with two amino acids (GLU199 and TRP279), creating two linkages (1 H-donor and 1 H-pi). The commercial inhibitor (**37**) bound to 1EEA via an interaction with one amino acid (TYR121), generating one H-pi linkage.

### 2.5. Lipinski’s Rule of Five and ADMET-Based Pharmacokinetics and Pharmacology

Lipinski’s Rule of Five has been wildly utilized for evaluating the drug-likeness of an inhibitor compound when considering its further development as a drug [54,64,65]. Lipinski’s Rule of Five includes “mass < 500 Da, LogP value < 5, H-donor linkages < 5, H-acceptors linkages < 10, and the molar refractivity values in the range of 40–130”. A compound satisfying ≥2 of these criteria is suggested to show drug-like properties [64,65]. The data are presented in Table 8. The identified major compounds **1**, **8**, **28**, **31**, **33**, **34**, and **35**, from the EPL extract and the commercial inhibitor (**11**) satisfied five of Lipinski’s rules, except for three compounds (**6**, **9**, and **11**) which satisfied three of the rules. The data indicate that all these compounds possess favorable drug-like properties and have a high potential for use as drugs. The predicted ADMET properties (Absorption, Distribution, Metabolism, Excretion, and Toxicity) of the above compounds and the commercial inhibitor (**37**) were also analyzed and the data are presented in the Appendix A section (Table A1 and Table A2). The ADMET analysis also gave positive results. In general, these identified compounds also showed good ADMET properties within the required limits [64].

## 3. Materials and Methods

### 3.1. Materials and Chemicals

The aerial parts of EPL were obtained from Yok Don National Park, Dak Lak Province, Vietnam in 2022. The identification of this medicinal plant was carried out by a botanist named Thi Thu Nguyen. The collected specimens were dried until a constant weight was reached. They were then packed in polyethylene bags and labeled as EPL-YD-2022 (*Eclipta prostrata* Linn—Yok Don—year 2022). These specimens were stored at 0–4 °C in the Natural Products Lab of the Institute of Biotechnology and Environment, Tay Nguyen University, until the extraction process.

The solvent for extraction was methanol (China). The chromatographic solvents, methanol and H3PO4, were purchased from Merk. Reagents including AChE; 2,2-diphenyl-1-picrylhydrazyl (DPPH); 2,2′-azino-bis(3-ethylbenzothiazoline-6-sulfonic acid) (ABTS); α-amylase; and α-Glucosidase were purchased from Sigma Aldrich.

### 3.2. Optimization of the Extraction Conditions of Flavonoid

#### 3.2.1. Preparation of EPL Extracts

Following the pressurized liquid extraction (PLE) technique described in a previous study [66], EPL powder (2.0 g) was treated in 20 mL of methanol, pressurized at 80–120 bar, at temperatures of 40–70 °C for 30–60 min using the PLE system E-916 (SpeedExtractor E-916, Buchi, Switzerland). The extract was dried using a rotary evaporator system (Laborota 4000, Heidolph, Germany) and stored at 0–4 °C for further investigations.

#### 3.2.2. Experimental Design

The extraction optimization was carried out using response surface methodology with help of Design Expert software version 11.0.0 (Stat ease Inc., Minneapolis, MN, USA). The test plan was performed by a three-factor/three-level design referred to as the Box–Behnken design (BBD). The experimental design included seventeen test runs, with five replications at the central mark. The central values were employed to obtain the test error and confirm the reproducibility property of the examined data. The independent variables in this research were temperature extraction (X_1_, °C), extraction period (X_2_, min) and pressure (X_3_, bar). The values of the independent variables are reported in Table 9 by their coded values as −1, 0, and +1 intervals, representing the low, medium, and large levels of each factor, respectively.

The regression analysis of the processes was executed on the experimental data of response in total flavonoid content (Y) as influenced by the extraction conditions and were fitted into the multinomial regression formula shown in Equation (2):(2)$$y=\beta_{0}+\overset{k}{\underset{i=1}{\sum }}\beta_{i}X_{i}+\overset{k}{\underset{i=1}{\sum }}\beta_{ii}X_{i}^{2}+\sum \underset{i<j}{\sum }\beta_{ij}X_{i}X_{j}$$
where Y corresponds to the response variable to be modeled; β_0_ is known as a constant, and βi, βii, and βij are linear, quadratic, and cross-product coefficients, respectively; Xi, and Xj are the orders of the independent variable; and k is the number of changes.

The optimal extracted conditions were chosen based on achieving the highest total flavonoids content using the desirability feature technique in the test Design Expert software version 11.0.0. The corresponding polynomial formula was described in the shape of 3D surface plots to depict the correlation between the response signals and the test changes applied. The ANOVA outcomes were determined using a 95.0% confidence interval, and this analysis was important to detect the best-fitting quadratic model for three factors. A regression equation was assessed by employing F statistics and the lack-of-fit test. According to the results, the sample is highly significant when the computed F-value is higher than the tabulated F-value and the probability result value is small (*p* < 0.00010). This indicates that the independent variables in a response sample have a significant impact on the interaction effect. Further parameters including the coefficient of correlation (R^2^), adjusted R^2^ and the coefficient of variance (CV%) are considered to confirm the model.

#### 3.2.3. Total Flavonoids

The total flavonoid content was determined using a colorimetric technique modified by Nguyen et al. [67]. The extract (0.010 mg/mL) was poured into a test tube containing 1.250 mL of distilled water. Thereafter, 0.0750 mL of 5.0% NaNO_2_ was added to the mixture, and incubated for 5.0 min. Next, 0.150 mL of 10.0% AlCl_3_ was added. After 6.0 min, 0.50 mL of a 1.0 M NaOH solution was added, and the compound mixture was diluted with 0.550 mL of distilled water. The absorbance of the resulting mixture was immediately measured at 510.0 nm. Flavonoid levels were calculated using the standard curve and illustrated as milligram quercetin equivalent (mg QE)/g of dry weight.

### 3.3. Chemical Analysis

#### 3.3.1. Ultra High-Performance Liquid Chromatography (UHPLC-UV) Method

The components in the EPL extracts were detected, identified, and quantified by the UHPLC system (Thermo Ultimate 3000, Waltham, MA, USA). The extracts were dissolved in MeOH at a concentration of 10.0 mg/mL and subsequently filtered using a Polyvinylidene difluoride membrane filter (0.45.0 μm, Millipore Sigma, Billerica, MA, USA) before injection. The separation of the sample constituents was achieved using a Hypersil GOLD aQ column (3 μm, 150 × 2.1 mm), which was maintained at a temperature of 30 °C. The mobile phase comprised methanol (MeOH) and water containing 0.10% H_3_PO_4_. The flow rate was maintained at 0.2 mL/min throughout the analysis, and the gradient program was set as follows: 0.0–0.5 min, 5% MeOH; 0.5–8.0 min, 5–30% MeOH; 8.0–13 min, 30–45% MeOH; 13.0–18.0 min, 45–65% MeOH; 18.0–22.0 min, 65–95% MeOH; and 22.0–26.0 min, 95–5% MeOH. The injection volume was 2 μL, and peaks were monitored simultaneously at 265 nm [54,68,69].

#### 3.3.2. Gas Chromatography-Mass Spectrometry (GC-MS) Method

The GC-MS system (GC-Thermo Trace GC Ultra, USA and ITQ 900 Mass Spec-Thermo, USA) was used to identify volatile chemicals in the EPL extracts. The plant extract was dissolved in methanol (MeOH) and subsequently purified using the QuEChERS method for solid-phase extraction [70,71]. The GC-MS analysis was conducted employing a TG-SQC capillary column (30 m × 0.25 mm × 0.25 μm). High purity helium (99.99%) was employed as the carrier gas with a constant flow rate of 1 mL/min. A sample solution of 1 μL was injected with a split ratio of 10:1. The injector temperature was set at 250 °C, while the ion-source temperature was maintained at 230 °C. The column was initially held at a temperature of 50 °C for 2 min, followed by a gradual increase to 250 °C at a rate of 10 °C/min. Subsequently, the temperature was further increased to 280 °C and held for 10 min. MS data were acquired with an electron energy of 70 eV, a scanning interval time of 0.5 s, and fragments ranging from 35 to 650 Da. Compound identification was carried out by comparing them with reported compounds using data from the Mass Spectra Library (NIST 17 and Wiley) [54,72,73].

### 3.4. Chemical Assays for Detection of In Vitro Anti-Oxidant, Anti-Diabetic, and Anti-Alzheimer Effects

The DPPH radical scavenging assay and the ABTS assay were used to assess anti-oxidant activity, as previously described in detail by Nguyen et al. [74]. In a 96-well plate, 100 μL of samples (at various concentrations) or distilled water (blank sample) were combined with 25 μL of 0.75 mM DPPH solution (dissolved in methanol). After that, the plates were kept in complete darkness for 30 min. Subsequently, the absorbance was determined at a wavelength of 517 nm. For the ABTS radical-scavenging activity assay, test samples (at various concentrations) were mixed with pre-diluted ABTS solution (reaching an optical density of 0.7). The reaction was allowed to occur for 10 min at room temperature. The absorbance was measured at 734 nm. Ascorbic acid was used as positive control.

Anti-glucosidase assays were conducted as previously described by Nguyen TH et al. [75]. Briefly, 50 μL of sample solution in 100 μL of potassium phosphate buffer was mixed with 50 μL of α-glucosidase solution. The mixed solution was incubated for 20 min at 37 °C. Then, 50 μL of p-nitrophenyl glucopyranoside was added and the mixture was incubated at 37 °C for 40 min. The absorbance of the solution containing p-nitrophenol was measured at a wavelength of 410 nm. The control sample using a commercial compound (acarbose) was tested under the same conditions for comparison purposes.

Anti-amylase assays were conducted as described by Nguyen et al. [76] and followed these steps. First, 50 μL of sample (at different concentrations) was mixed with 150 μL of α-amylase solution (0.25 U/mL) and incubated for 10 min at 37 °C. Then, we added 200 μL of soluble starch 0.25% and measured the optical density at 540nm after being kept for 20 min at 37 °C. For comparison, acarbose was used as a standard inhibitor of the enzyme.

The anti-Alzheimer impact was assessed in vitro utilizing the key enzyme acetylcholinesterase inhibition assay, as described in an earlier study [77]. In detail, 120 μL of phosphate buffer, 60 μL of tested sample, and 60 μL of enzymatic solution (0.5 mM) were mixed and incubated for fifteen minutes at 25 °C in a 96-well plate. This was followed by the addition of 30 μL of 5,5′-dithiobis-2-nitrobenzoic acid (0.003 M) and 40 μL of acetylthiocholine iodide (0.002 M). The reaction mixture was kept in an incubator at 25 °C for ten minutes. The absorbance of the solution was measured at 415 nm. Berberine chloride was used as a standard inhibitor of the enzyme.

All chemical assays were conducted in triplicate. The inhibition of enzyme activity was calculated by the following formula:Inhibitory activity (%) = (A − B)/A × 100%.
where A is the average absorbance of a blank sample and B is the average absorbance of the tested samples. The IC_50_ value (half-maximal inhibitory concentration of samples) was determined via a standard equation constructed by the inhibitory activity (%) and the respective concentration of the tested sample.

### 3.5. Virtual Study Method

The molecular docking study was performed following the protocol presented in previous reports [54,65]. The protein structure data of the enzyme 1EEA were obtained from the Worldwide Protein Data Bank (https://www.rcsb.org/structure/3qpc, accessed on 7 August 2023) for the preparation of the 3D structure by using MOE-2015.10 software. The 3D structures of ligands (major compounds **1**, **6**, **8**, **9**, **11, 28**, **31**, **33**, **34**, and **35**) and berberine chloride (**37**) were prepared and optimized using MOE-2015.10. The ligands were docked into the active binding site on protein 1EEA using the same software. The output data, such as RMSD, DS, types of interaction, amino acid composition, and the distances of linkages, were obtained for analysis.

The drug-likeness was analyzed via the Lipinski’s Rules of Five method using the online software (http://www.scfbioiitd.res.in/software/drugdesign/lipinski.jsp, accessed on 7 August 2023), and a web tool (http://biosig.unimelb.edu.au/pkcsm/theory, accessed on 7 August 2023) was utilized to predict pharmacokinetic parameters including absorption, distribution, metabolism, excretion and toxicity (ADMET).

### 3.6. Statistical Analysis

Each assay was repeated three times and the results were expressed as mean ± standard deviation (SD). A statistical analysis is reported for the results of all tested cases and a 95.0% confidence interval value (*p* < 0.05) was used.

## 4. Conclusions

In summary, this study successfully identified the optimal conditions for extracting the highest yield of total flavonoid content from *Eclipta prostrata* L using PLE. The findings suggest that PLE outperforms other extraction methods in terms of efficiency. Additionally, the application of PLE to extract TFC compounds from EPL collected in Tay Nguyen and the emphasis on environmentally conscious practices contribute to the uniqueness and significance of this research. Moreover, the identification of some compounds with high TFC, such as wedelolactone (**1**), chlorogenic acid (**3**), epicatechin gallate (**6**), salicylic acid (**8**), isovitexin (**9**), apigetrin (**11**), and myricetin (**12**), may contribute to explaining the bioactivities of EPL.

The results of the bioassays align with the traditional use of EPL for enhancing memory and cognitive function, as well as its potential benefits in diabetes management due to its α-amylase and α-glucosidase inhibition. The evaluation of the extract’s abilities to act as an anti-oxidant and inhibit enzymes like α-amylase, α-glucosidase, and acetylcholinesterase provides valuable insights into its potential therapeutic effects, especially in conditions like Alzheimer’s disease and diabetes. Furthermore, a virtual study was conducted, and the results indicated that the major compounds (**1**, **6**, **9**, and **11**) demonstrated a more effective inhibition against the target enzymes than berberine chloride, with good binding energy (DS values, −12.3 to −14.3 kcal/mol) and acceptable values of RMSD (1.02–1.67 Å), and most of these major constituents exhibited good ADMET properties in the required allotted limitation.

## Figures and Tables

**Figure 1 pharmaceuticals-16-01476-f001:**
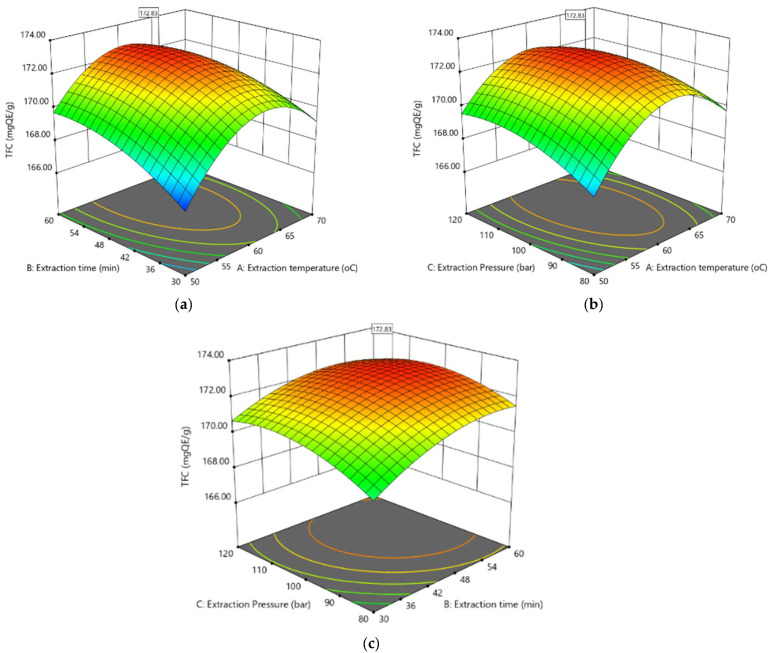
The response surface zone plot of total flavonoid content: (**a**) extraction temperature and extraction time, (**b**) extraction temperature and extraction pressure, and (**c**) extraction time and extraction pressure.

**Figure 2 pharmaceuticals-16-01476-f002:**
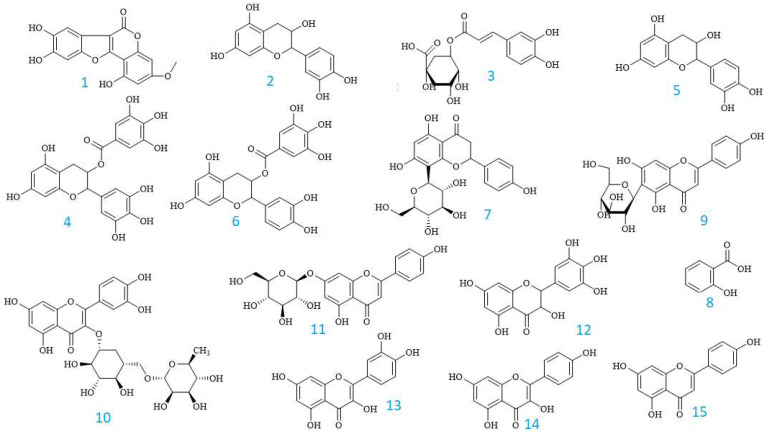
The chemical structures of compounds identified from the EPL extracts by UHPLC analysis.

**Figure 3 pharmaceuticals-16-01476-f003:**
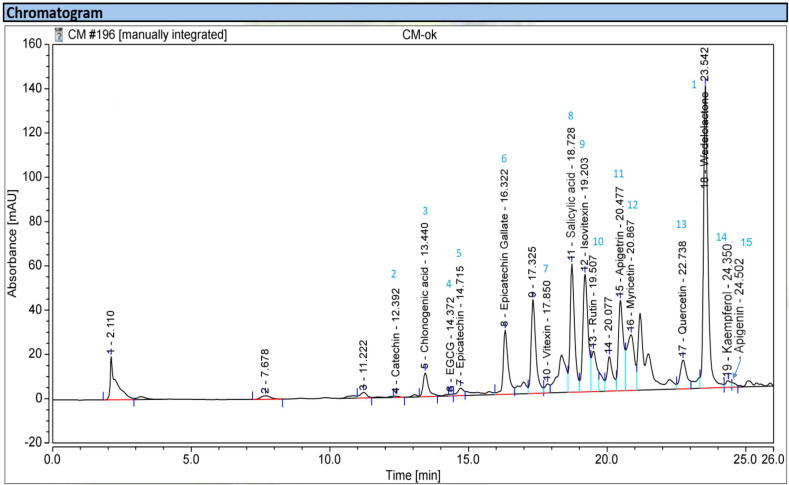
High-performance liquid chromatography fingerprints of the MeOH extract of EPL.

**Figure 4 pharmaceuticals-16-01476-f004:**
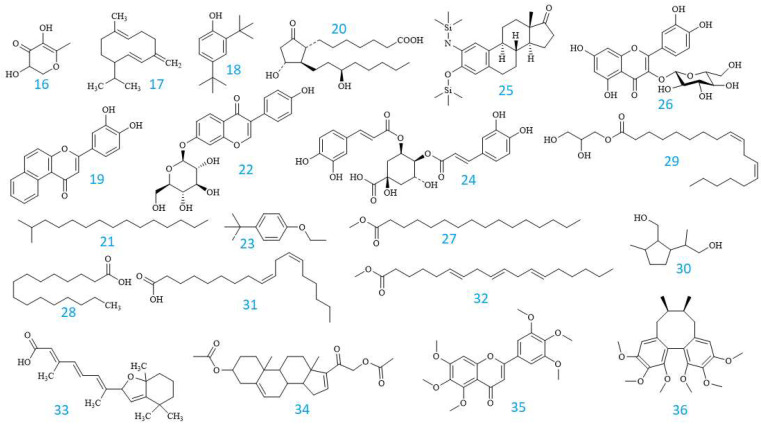
GC profile of volatile compounds identified from the MeOH extract of EPL.

**Figure 5 pharmaceuticals-16-01476-f005:**
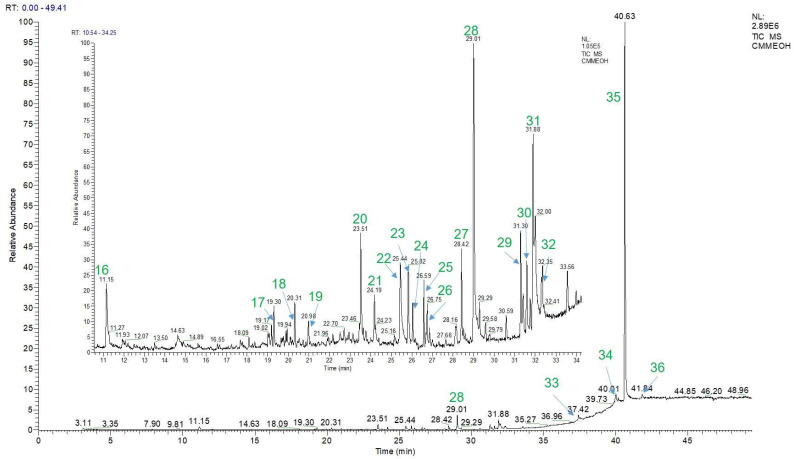
GC-MS fingerprints of the MeOH extract of EPL.

**Figure 6 pharmaceuticals-16-01476-f006:**
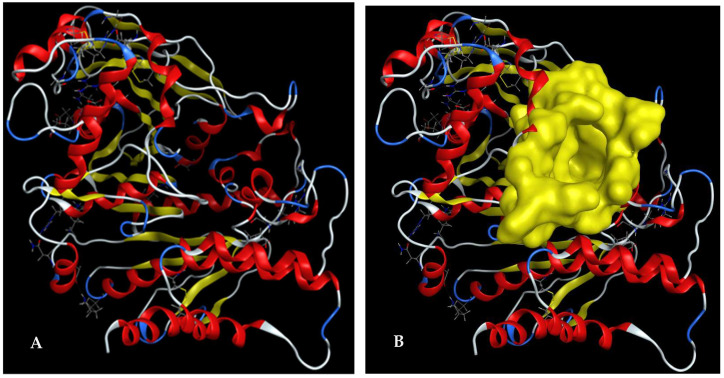
The 3D structure of Acetylcholinesterase (**A**), and the 3D structure of the binding site on Acetylcholinesterase determined using the site finder function of MOE (**B**).

**Figure 7 pharmaceuticals-16-01476-f007:**
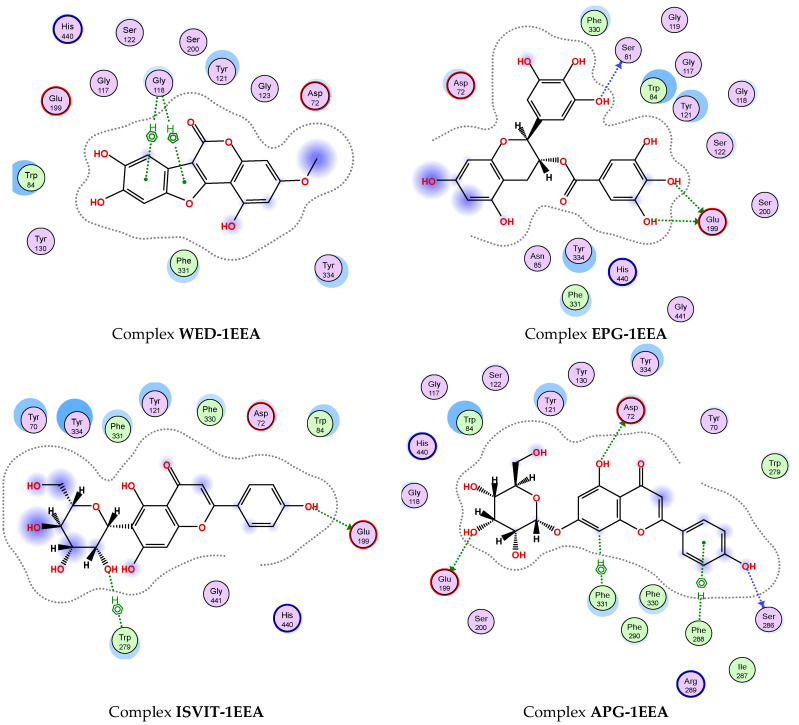
The interaction of active compounds (**1**, **6**, **9**, and **11**) and berberine chloride (**37**) with acetylcholinesterase (1EEA) at the biding site on 1EEA.

**Table 1 pharmaceuticals-16-01476-t001:** The test data were acquired for the three responses according to the BBK matrix.

Run No	Extraction Temperature	Extraction Time	Extraction Pressure	Total Flavonoid Content
	X_1_ (°C)	X_2_ (min)	X_3_ (bar)	Y_1_ (mg QE/g)
1	50	45	120	169.28
2	50	45	80	167.05
3	60	45	100	172.63
4	60	60	120	171.97
5	50	30	100	166.72
6	60	45	100	171.85
7	70	45	80	169.48
8	70	60	100	170.23
9	60	30	80	169.24
10	60	60	80	171.52
11	50	60	100	169.63
12	60	30	120	170.59
13	60	45	100	173.05
14	70	45	120	169.81
15	60	45	100	172.39
16	60	45	100	172.51
17	70	30	100	169.12

**Table 2 pharmaceuticals-16-01476-t002:** Regression coefficients of the predicted second-order polynomial models for the total flavonoid content.

Source	Total Flavonoid Content	
	F-Value	*p*-Value
Model	51.97	<0.0001 ^S^
X_1_	36.93	0.0005 ^S^
X_2_	61.31	0.0001 ^S^
X_3_	19.76	0.0030 ^S^
X_1_X_2_	6.74	0.0357 ^S^
X_1_X_3_	7.51	0.0289 ^S^
X_2_X_3_	1.68	0.2355 ^NS^
X_1_^2^	263.46	<0.0001 ^S^
X_2_^2^	23.43	0.0019 ^S^
X_3_^2^	24.59	0.0016 ^S^
Lack of Fit	0.1569	0.9200 ^NS^
R^2^	0.9853	
Adjusted R^2^	0.9663	
C.V%	0.2035	

S: significant (*p* < 0.05); NS: non-significant.

**Table 3 pharmaceuticals-16-01476-t003:** The content of phenolic compounds in the MeOH extract from EPL.

No.	Compound	Content (μg/g of Dried Extract)
1	Wedelolactone	1652.93 ± 0.04
2	Catechin	44.52 ± 0.03
3	Chlorogenic Acid	420.54 ± 0.03
4	EpiGalloCatechin Gallate	9.23 ± 0.02
5	Epicatechin	257.61 ± 0.03
6	Epicatechin gallate	537.78 ± 0.02
7	Vitexin	58.69 ± 0.02
8	Salicylic acid	7912.08 ± 0.03
9	Isovitexin	537.49 ± 0.04
10	Rutin	313.51 ± 0.03
11	Apigetrin	519.70 ± 0.03
12	Myricetin	465.42 ± 0.03
13	Quercetin	141.49 ± 0.04
14	Kaempferol	33.72 ± 0.01
15	Apigenin	7.23 ± 0.01

**Table 4 pharmaceuticals-16-01476-t004:** Profile of volatile compounds of EPL detected by GC-MS.

No.	RT	% Area	Name
16	11.15	0.89 ± 0.01	2,3-Dihydro-3,5-dihydroxy-6-methyl-4H-pyran-4-one
17	19.30	0.33 ± 0.01	Germacrene D
18	20.31	0.35 ± 0.01	Phenol, 2,4-bis(1,1-dimethylethyl)-
19	20.98	0.22 ± 0.01	3’,4’-Dihydroxy-β-naphthoflavone
20	23.51	1.18 ± 0.02	13,14-Dihydro-15(R)-prostaglandin E1
21	24.19	0.57 ± 0.02	Pentadecane, 2-methyl-
22	25.44	1.25 ± 0.05	Daidzin
23	25.82	0.77 ± 0.02	Benzene, 1-(1,1-dimethylethyl)-4-ethoxy-
24	26.05	0.26 ± 0.01	4,5-Dicaffeoylquinic acid
25	26.59	0.40 ± 0.02	Estra-1,3,5(10)-trien-17-one, 2-[(trimethylsilyl)amino]-3-[(trimethylsilyl)oxy]-
26	26.75	0.38 ± 0.04	Isoquercitin
27	28.42	0.77 ± 0.02	Hexadecanoic acid, methyl ester
28	29.01	3.01 ± 0.02	Hexadecanoic acid
29	31.30	0.69 ± 0.01	9,12-Octadecadienoic acid (Z,Z)-, 2,3-dihydroxypropyl ester
30	31.59	0.55 ± 0.02	Cyclopentaneethanol, 2-(hydroxymethyl)-β,3-dimethyl-
31	31.88	3.59 ± 0.04	9,12-Octadecadienoic acid (Z,Z)-
32	32.35	0.71 ± 0.05	6,9,12-Octadecatrienoic acid, methyl ester
33	37.42	2.16 ± 0.04	all-trans-5,8-Epoxyretinoic acid
34	40.01	2.33 ± 0.02	Pregna-5,16-dien-20-one, 3,21-bis(acetyloxy)-, (3β)-
35	40.63	77.97 ± 0.05	3′,4′,5′,5,6,7-hexamethoxyflavone
36	41.84	0.77 ± 0.04	Schizandrin A

**Table 5 pharmaceuticals-16-01476-t005:** The bioactivities of EPL.

Sample	AChE IC_50_ (μg/mL)	DPPH IC_50_ (μg/mL)	ABTS IC_50_ (μg/mL)	α-amylase IC_50_ (μg/mL)	α-Glucosidase IC_50_ (μg/mL)
EPL	76.8 ± 0.8	130.2 ± 1.0	79.9 ± 0.9	429.9 ± 2.1	223.1 ± 1.3
Berberine chloride	301.1 ± 1.2	Nd	Nd	Nd	Nd
Vitamin C	Nd	38.4 ± 0.5	23.0 ± 0.4	Nd	Nd
Salicylic acid	55.98 ± 0.12	Nd	Nd	Nd	Nd
Wedelolactone	61.23 ± 0.21	Nd	Nd	Nd	Nd
Acarbose	Nd	Nd	Nd	24.9 ± 0.3	9.3 ± 0.1

Nd: not determined.

**Table 6 pharmaceuticals-16-01476-t006:** The IC_50_ values of some phenolic compounds in the MeOH extract from EPL.

No.	Compound	Content (μg/g of Dried Extract)	Reference
3	Chlorogenic Acid	98.17	[56]
4	EpiGalloCatechin Gallate	16.83 ± 0.12	[57]
7	Vitexin	12.16 ± 3.58	[58]
9	Isovitexin	6.24 ± 1.15	[58]
10	Rutin	0.219 ± 0.011	[59]
12	Myricetin	3.95 ± 0.61	[57]
13	Quercetin	0.181 ± 0.023 to 9.56 ± 0.37	[57,58,59]
14	Kaempferol	3.05 ± 0.77	[57]
15	Apigenin	7.72 ± 0.15 to 34.43 ± 2.41	[57,58]

**Table 7 pharmaceuticals-16-01476-t007:** The docking study results of ligands (10 compounds) and berberine chloride binding with acetylcholinesterase (1EEA).

Ligands	Symbol of Complex	RMSD (Å)	DS (kcal/mol)	Linkages	Amino Acids Interacting with the Ligands (Distance (Å)/E (kcal/mol)/ Linkage Type)
Wedelolactone (**1**)	WED-1EEA-1(**1**)	1.02	−12.3	2 linkages (2 pi-H)	GLY118 (4.37/-0.6/pi-H) GLY118 (3.70/-0.6/pi-H)
Epicatechin gallate (**6**)	EPG-1EEA-1(**17**)	1.45	−14.3	3 linkages (3 H-donor)	SER81 (2.68/-2.4/H-donor) GLU199 (3.24/-2.1/H-donor) GLU199 (2.68/-5.6/H-donor)
Salicylic acid (**8**)	SALA-1EEA-1(**25**)	1.98	−10.2	3 linkages (1 H-donor, 1 H-pi, 1 pi-pi)	SER122 (2.90/-1.9/H-donor) PHE330 (4.28/-0.6 H-pi) TRP84 (3.97/-0.0 pi-pi)
Isovitexin (**9**)	ISVIT-1EEA-1(**37**)	1.67	−13.4	2 linkages (1 H-donor, 1 H-pi)	GLU199 (3.14/-1.6 H-donor) TRP279 (4.49/-0.9 H-pi)
Apigetrin (**11**)	APG-1EEA-1(**42**)	1.44	−13.7	5 linkages (3 H-donor, 1 H-pi, 1 pi-H)	GLU199 (2.93/-2.8 H-donor) ASP72 (3.01/-1.2 H-donor) SER 286 (2.98/-0.8 H-donor) PHE331 (4.57/-0.6 H-pi) PHE288 (4.69/-0.7 pi-H)
Hexadecanoic acid (**28**)	HDA-1EEA-1(**156**)	1.81	−11.7	3 linkages (1 H-donor, 1 H-acceptor, 1 pi-H)	SER286 (2.96/-1.9 H-donor) ARG289 (3.21/-1.9 H-acceptor) TRP279 (4.58/-0.7 H-pi)
9,12-Octadecadienoic acid (Z,Z)- (**31**)	ODDA-1EEA-1(**126**)	1.59	−11.5	2 linkages (1 H-donor, 1 H-pi)	TYR70 (2.84/-3.9 H-donor) TYR334 (4.14/-0.8 H-pi)
all-trans-5,8-Epoxyretinoic acid (**33**)	EPRA-1EEA-1(**79**)	1.11	−10.6	1 linkage (1 H-acceptor)	TYR121 (3.25/-0.7 H-acceptor)
Pregna-5,16-dien-20-one, 3,21-bis(acetyloxy)-, (3β)- (**34**)	PREGD-1EEA-1(**81**)	1.42	−10.4	1 linkage (1 H-acceptor)	ARG289 (3.42/-1.0 H-acceptor)
3′,4′,5′,5,6,7-hexamethoxyflavone (**35**)	HMF-1EEA-1(**97**)	1.75	−10.0	1 linkage (1 H-pi)	PHE330 (4.46/-0.6 H-pi)
Berberine chloride (**37**)	BC-1EEA1(**3**)	1.65	−12.1	1 linkage (1 H-pi)	TYR121 (4.50/-0.7 H-pi)

**Note**: Wedelolactone (**WED**), Epicatechin gallate (**EPG**), Salicylic acid (**SALA**), Isovitexin (**ISVIT**), Apigetrin (**APG**), Hexadecanoic acid (**HDA**), 9,12-Octadecadienoic acid (Z,Z)- **(ODDA)**, all-trans-5,8-Epoxyretinoic acid **(EPRA)**, Pregna-5,16-dien-20-one, 3,21-bis(acetyloxy)-, (3β)- **(PREGD)**, 3′,4′,5′,5,6,7-hexamethoxyflavone **(HMF)**, berberine chloride (**BC**).

**Table 8 pharmaceuticals-16-01476-t008:** Lipinski’s Rule of Five analysis of compounds from *Eclipta prostrata* L.

Compound	Mass (Dalton)	Hydrogen Bond Donor	Hydrogen Bond Acceptors	LogP	Molar Refractivity
Wedelolactone (**1**)	314.0	3	7	2.758	78.15
Epicatechin gallate (**6**)	458.0	8	11	2.233	108.92
Salicylic acid (**8**)	137.0	1	3	−0.244	32.44
Isovitexin (**9**)	432.0	7	10	−0.066	103.53
Apigetrin (**11**)	432.0	6	10	−0.107	103.54
Hexadecanoic acid (**28**)	255.0	0	2	4.218	75.32
9,12-Octadecadienoic acid (Z,Z)- (**31**)	279.0	0	2	4.550	84.36
all-trans-5,8-Epoxyretinoic acid (**33**)	317.0	0	3	3.330	89.33
Pregna-5,16-dien-20-one, 3,21-bis(acetyloxy)-, (3β)- (**34**)	414.0	0	5	4.549	112.36
3′,4′,5′,5,6,7-hexamethoxyflavone (**35**)	402.0	0	8	3.354	105.13
Berberine chloride (**37**)	337.0	0	3	2.733	93.03
Five Lipinski Rules	<500	<5	<10	<5	40–130

**Table 9 pharmaceuticals-16-01476-t009:** Extraction variables selected for total flavonoid content optimization.

Ranges of Independent Variable	Independent Variable			Dependent Variable (Y)
	Extraction Temperature (X_1_-^o^C)	Extraction Time (X_2_-min)	Extraction Pressure (X_3_-bar)	
−1	50	30	80	Total flavonoid content
0	60	45	100	
1	70	60	120	

## Data Availability

The data used to support the findings of this study are included in the article.

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
