# Peer review of "Assessment of the Chemical Profile and Potential Medical Effects of a Flavonoid-Rich Extract of Eclipta prostrata L. Collected in the Central Highlands of Vietnam"

_pharmaceuticals, 2023, doi:10.3390/ph16101476_

Round 1
Reviewer 1 Report
The article complies with the aims and scope of this journal. please revise again the entire document in the search for some typos and overall English grammar and syntax. Also, If it would be possible to increase the quality of the figures containing/describing the chromatograms since they seem more like a screenshot from the equipment than a proper figure. An article so complete as this one must have high quality images.
The article complies with the aims and scope of this journal. please revise again the entire document in the search for some typos and overall English grammar and syntax.
Author Response
Reviewer # 1
My comments regarding the manuscript entitled, “Assessment of Chemical Profile and Potential Medical Effect of Eclipta prostrata L. Extract Rich in Flavonoids Collected in The Central Highland of Vietnam", are as follows:
- The article complies with the aims and scope of this journal. please revise again the entire document in the search for some typos and overall English grammar and syntax. Also, If it would be possible to increase the quality of the figures containing/describing the chromatograms since they seem more like a screenshot from the equipment than a proper figure. An article so complete as this one must have high quality images.
Reply: The English writing of this manuscript was checked and carefully revised. Some figures were updated for better quality. We pay our respects and thanks to your positive and important comments for enhancing the quality of this paper.
Dear Reviewer/Advancer
We feel pleasure to thank you for your time and effort, as well as your excellent suggestions for refining the readability and impact of the manuscript. We have gone through all the suggestions cautiously and made the revisions accordingly, and all amended parts have been typed in red in the revised manuscript. Finally, we like to express our deep thanks for your comments and suggestions again. You certainly have served to improve the quality of this paper. We hope our response is satisfactory.
Looking forward to hearing from you.
Thanking you,
Yours Sincerely,
San-Lang Wang
Reviewer 2 Report
DPPH radical scavenging, ABTS, Anti-glucosidase, and Anti-amylase assays are not biological assays, are chemical assays.
Author Response
Reviewer # 2
DPPH radical scavenging, ABTS, Anti-glucosidase, and Anti-amylase assays are not biological assays, are chemical assays.
Reply: Thanks for your detection. All these mistakes was edited in Revised version manuscript.
Dear Reviewer/Advancer
We feel pleasure to thank you for your time and effort, as well as your excellent suggestions for refining the readability and impact of the manuscript. We have gone through all the suggestions cautiously and made the revisions accordingly, and all amended parts have been typed in red in the revised manuscript. Finally, we like to express our deep thanks for your comments and suggestions again. You certainly have served to improve the quality of this paper. We hope our response is satisfactory.
Looking forward to hearing from you.
Thanking you,
Yours Sincerely,
San-Lang Wang
Reviewer 3 Report
The paper describes in detail the optimization of extraction procedure with respect to maximizing the content of flavonoids. Then, the biological activity of the extract is assayed in a couple of ways. Most emphasis is on acetylcholine esterase inhibition activity. The actual assay compares the extract with berberine, and the extract is ca 4-times more potent in inhibition than berberine. Then there is a big modeling portion, which is not convincing. There is a definite need to test inhibition activity of major components of the extract in parallel to decide which component is actually so active. Docking energies are not that different for all components, despite their chemical structure is quite different, and docking doesn't explain the actual data.
In my opinion, the experimental tests of inhibition activity are necessary to be done for wedelolactone and salycyllic acid to judge on the source of high activity in the extract.
Flavonoids are known to trigger antioxidant defense program, and frankly speaking, ARE-luciferase assay could be done to compare the extract with major components identified. ROS quenching is not the actual role of flavonoids. Plus, flavonoids activate estrogen receptor and trigger antioxidant and anti-inflammatory response via this pathway.
English is quite good, some light polishing may be necessary.
Author Response
Reviewer # 3
The paper describes in detail the optimization of extraction procedure with respect to maximizing the content of flavonoids. Then, the biological activity of the extract is assayed in a couple of ways. Most emphasis is on acetylcholine esterase inhibition activity. The actual assay compares the extract with berberine, and the extract is ca 4-times more potent in inhibition than berberine. Then there is a big modeling portion, which is not convincing. There is a definite need to test inhibition activity of major components of the extract in parallel to decide which component is actually so active. Docking energies are not that different for all components, despite their chemical structure is quite different, and docking doesn't explain the actual data.
In my opinion, the experimental tests of inhibition activity are necessary to be done for wedelolactone and salycyllic acid to judge on the source of high activity in the extract.
Flavonoids are known to trigger antioxidant defense program, and frankly speaking, ARE-luciferase assay could be done to compare the extract with major components identified. ROS quenching is not the actual role of flavonoids. Plus, flavonoids activate estrogen receptor and trigger antioxidant and anti-inflammatory response via this pathway.
Reply:
Firstly, we would like to express our sincere gratitude for your meticulous review and the valuable comments provided to enhance the quality of our manuscript.
As per your suggestion, we conducted inhibition activity tests for wedelolactone and salicylic acid. The results showed that salicylic acid and wedelolactone exhibited more effective AChE inhibitory activity, with IC50 values of 55.98 ± 0.12 µg/mL and 61.23 ± 0.21 µg/mL, respectively, compared to the IC50 value of 76.8 ± 0.8 µg/mL for the EPL extract (newly added in Table 5). These results suggest that salicylic acid and wedelolactone play a significant role in the anti-AChE activity of the EPL extract. Additionally, we included the IC50 values of other components in the EPL extract, citing previous research references (newly added in Table 6). Notably,
many of the flavonoids present in the EPL extract demonstrated high AChE inhibition potential. This is the reason why the extract is ca 4-times more potent in inhibition than berberine.
Regarding the decision not to conduct the ARE-luciferase assay in this research, our study was primarily a screening study aimed at assessing fundamental activities. Research has indicated that oxidative stress, diabetes, and Alzheimer's disease are interconnected. Non-communicable diseases such as diabetes and Alzheimer's are posing a growing health challenge globally and in Vietnam. Hence, our study focused on evaluating the potential activities of this extract in addressing these health concerns. The results demonstrated that the extract holds promise in Alzheimer's disease treatment, which prompted further investigations through molecular docking to gain a better understanding of the substances' roles. However, to make this extract applicable to Alzheimer's disease treatment, we recognize the need for next research, as suggested by the reviewer. These investigations may include assessing the ability to inhibit Alzheimer's-related enzymes, anti-inflammatory, and antioxidation through ARE-luciferase assays and other relevant studies.
Dear Reviewer/Advancer
We feel pleasure to thank you for your time and effort, as well as your excellent suggestions for refining the readability and impact of the manuscript. We have gone through all the suggestions cautiously and made the revisions accordingly, and all amended parts have been typed in red in the revised manuscript. Finally, we like to express our deep thanks for your comments and suggestions again. You certainly have served to improve the quality of this paper. We hope our response is satisfactory.
Looking forward to hearing from you.
Thanking you,
Yours Sincerely,
San-Lang Wang
Reviewer 4 Report
The manuscript from Phan et al reports optimization of flavonol extraction from the traditional medicinal plant Eclipta prostrata using pressurized liquid extraction. Compounds present in the extract were identified by UHPLC-UV and GC-MS. In silico docking studies of the more abundant compounds to acetyl cholinesterase suggested that some of these compounds may be responsible for the effects seen when the plant is used as a traditional medicine.
The science appears to be carefully performed and clearly reported. Minor improvements to the English are needed and some suggestions are listed below.
line 27 replace "acid salicylic" with "salicylic acid"
line 51 replace "enhance" with "and to enhance"
linr 53 replace "on isolated and characterized" with "to isolate and characterize"
line 54 replace "determined" with "determine"
line 60 italicize "E. prostrata"
line 62 delete "something else"
line 68 replace "Beside" with "Besides"
line 136 define "QE"
line 143 replace "heat extraction" with "extraction temperature"
line 163 replace "certain" with "constant"
line 180 replace "with 53 min and" with "for 53 min at"
line 180 delete "to"
line 213 replace "analysis" with "analysed"
line 222 replace "inhibit" with "inhibits"
line 223 replace "inhibitory" with "and shows "inhibitory"
line 234 replace "has been" with "that has been"
line 260 does the "2-methyl" refer to the "Pentadecane" or is something missing?
line 282 define "DPPH" and "ABTS"
line 307 replace "anti-" with "oppose"
line 318 replace "evaluated" with "evaluate"
line 325 italicize "Eclipta prostrata"
line 328 define "ACh"
line 330 delete "activities"
line 338 replace "The most" with "A"
line 338 replace "determined" with "identified"
lines 391 and 392 replace "highest inhibitory effect" with "best docking score". Inhibitory activity was measured, only docking score.
line 399 replace "HE331" with "PHE331"
line 399 replace "lingkages" with "linkages"
line 413 replace "drug" with "a drug"
line 416 replace "likeness" with "like"
line 421 replace "ADMET" with "predicted ADMET"
line 425 What does "required allotted limitation" mean?
line 470 delete "purified". These compounds were identified but not purified.
line 476 define "PE"
line 516 replace "analyzed" with "analysis"
lines 528,529,530 Should be rewritten in the past tense.
line 540 delete one of the extra "."
line 549 Provide a reference for the "QuEChERS" method.
line 558 replace "Ingredient" with "Compound"
Comments on the English are listed in the comments box above.
Author Response
Reviewer # 4
The manuscript from Phan et al reports optimization of flavonol extraction from the traditional medicinal plant Eclipta prostrata using pressurized liquid extraction. Compounds present in the extract were identified by UHPLC-UV and GC-MS. In silico docking studies of the more abundant compounds to acetyl cholinesterase suggested that some of these compounds may be responsible for the effects seen when the plant is used as a traditional medicine.
The science appears to be carefully performed and clearly reported. Minor improvements to the English are needed and some suggestions are listed below.
line 27 replace "acid salicylic" with "salicylic acid"
line 51 replace "enhance" with "and to enhance"
linr 53 replace "on isolated and characterized" with "to isolate and characterize"
line 54 replace "determined" with "determine"
line 60 italicize "E. prostrata"
line 62 delete "something else"
line 68 replace "Beside" with "Besides"
line 136 define "QE"
line 143 replace "heat extraction" with "extraction temperature"
line 163 replace "certain" with "constant"
line 180 replace "with 53 min and" with "for 53 min at"
line 180 delete "to"
line 213 replace "analysis" with "analysed"
line 222 replace "inhibit" with "inhibits"
line 223 replace "inhibitory" with "and shows "inhibitory"
line 234 replace "has been" with "that has been"
line 260 does the "2-methyl" refer to the "Pentadecane" or is something missing?
line 282 define "DPPH" and "ABTS"
line 307 replace "anti-" with "oppose"
line 318 replace "evaluated" with "evaluate"
line 325 italicize "Eclipta prostrata"
line 328 define "ACh"
line 330 delete "activities"
line 338 replace "The most" with "A"
line 338 replace "determined" with "identified"
lines 391 and 392 replace "highest inhibitory effect" with "best docking score". Inhibitory activity was measured, only docking score.
line 399 replace "HE331" with "PHE331"
line 399 replace "lingkages" with "linkages"
line 413 replace "drug" with "a drug"
line 416 replace "likeness" with "like"
line 421 replace "ADMET" with "predicted ADMET"
line 425 What does "required allotted limitation" mean?
line 470 delete "purified". These compounds were identified but not purified.
line 476 define "PE"
line 516 replace "analyzed" with "analysis"
lines 528,529,530 Should be rewritten in the past tense.
line 540 delete one of the extra "."
line 549 Provide a reference for the "QuEChERS" method.
line 558 replace "Ingredient" with "Compound"
Reply: Thanks for your detection. All these mistake was revised as your suggestion.
Dear Reviewer/Advancer
We feel pleasure to thank you for your time and effort, as well as your excellent suggestions for refining the readability and impact of the manuscript. We have gone through all the suggestions cautiously and made the revisions accordingly, and all amended parts have been typed in red in the revised manuscript. Finally, we like to express our deep thanks for your comments and suggestions again. You certainly have served to improve the quality of this paper. We hope our response is satisfactory.
Looking forward to hearing from you.
Thanking you,
Yours Sincerely,
San-Lang Wang
Round 2
Reviewer 3 Report
The authors assayed the inhibitory activity of 2 major components of the extract, showing that they are likely to be the reason for high inhibitory activity of the extract itself with respect to acetylcholinestaerase.
I recommend accepting as is.